# Deciphering Gut Microbiome Responses upon Microplastic Exposure via Integrating Metagenomics and Activity-Based Metabolomics

**DOI:** 10.3390/metabo13040530

**Published:** 2023-04-07

**Authors:** Pengcheng Tu, Jingchuan Xue, Huixia Niu, Qiong Tang, Zhe Mo, Xiaodong Zheng, Lizhi Wu, Zhijian Chen, Yanpeng Cai, Xiaofeng Wang

**Affiliations:** 1Department of Environmental Health, Zhejiang Provincial Center for Disease Control and Prevention, 3399 Binsheng Road, Hangzhou 310051, China; 2Guangdong Provincial Key Laboratory of Water Quality Improvement and Ecological Restoration for Watersheds, Institute of Environmental and Ecological Engineering, Guangdong University of Technology, Guangzhou 510006, China; 3School of Medicine, Ningbo University, Ningbo 315000, China; 4College of Standardization, China Jiliang University, Hangzhou 310018, China; 5Department of Food Science and Nutrition, Zhejiang University, Hangzhou 310058, China

**Keywords:** microplastic exposure, gut microbiome, metabolomics, toxicity

## Abstract

Perturbations of the gut microbiome are often intertwined with the onset and development of diverse metabolic diseases. It has been suggested that gut microbiome perturbation could be a potential mechanism through which environmental chemical exposure induces or exacerbates human diseases. Microplastic pollution, an emerging environmental issue, has received ever increasing attention in recent years. However, interactions between microplastic exposure and the gut microbiota remain elusive. This study aimed to decipher the responses of the gut microbiome upon microplastic polystyrene (MP) exposure by integrating 16S rRNA high-throughput sequencing with metabolomic profiling techniques using a C57BL/6 mouse model. The results indicated that MP exposure significantly perturbed aspects of the gut microbiota, including its composition, diversity, and functional pathways that are involved in xenobiotic metabolism. A distinct metabolite profile was observed in mice with MP exposure, which probably resulted from changes in gut bacterial composition. Specifically, untargeted metabolomics revealed that levels of metabolites associated with cholesterol metabolism, primary and secondary bile acid biosynthesis, and taurine and hypotaurine metabolism were changed significantly. Targeted approaches indicated significant perturbation with respect to the levels of short-chain fatty acids derived from the gut microbiota. This study can provide evidence for the missing link in understanding the mechanisms behind the toxic effects of microplastics.

## 1. Introduction

It has been well demonstrated that the gut microbiota plays a key role in immune response [1,2], metabolic processes [3], epithelial homeostasis [4], etc. Mounting evidence has established significant associations between an imbalanced gut microbiota and various adverse health outcomes such as inflammation [5], obesity [6], diabetes [7,8], and cancer [9]. The gut microbiota evolves through several transitions during infancy and relatively stabilizes thereafter if no significant perturbations occur. However, it has been well recognized that exposure to numerous environmental chemicals, including heavy metals [10,11,12,13,14], pesticides [14,15], artificial sweeteners [15,16,17], and others [18], is able to alter the gut microbiome. In particular, environmentally driven perturbations in the gut microbiome may lead to adverse effects on the host. For example, arsenic exposure perturbs the gut microbiome and induces alterations of diverse microbiota-related metabolites, which is suggested as a novel mechanism underlying arsenic toxicity [13,19,20]. Likewise, nicotine-induced changes in gut microbial metabolic pathways and metabolites related to neurotransmitters might contribute to the neurotoxicity of nicotine [21]. The functional damage driven by environmental exposure in the gut microbiome has recently been proposed as gut microbiome toxicity [18].

Recently, microplastic pollution and its potential health effects have become an emerging global environmental health issue. Plastic particles with a size of less than 5 mm are usually called microplastics [2]. Widespread occurrence of microplastics has been reported in a variety of environmental matrices, including oceans, rivers, soil, and even table salt [1]. Microplastics can easily accumulate in the environment due to their ubiquity and persistence. Studies have shown that microplastics can be ingested by marine organisms and passed through the food chain [7,8]. Microplastics have been detected in a variety of food products such as bottled water, honey, beer, and canned fish [22]. Thus, similar to how humans can be exposed to organic contaminants such as bisphenol A, they can also be exposed to microplastics via a variety of exposure routes [23,24]. The resulting accumulation of microplastics in tissues may cause a variety of toxic effects, such as growth inhibition, energy deficiency, inflammation, oxidative stress, and metabolic abnormalities [7,8]. However, the potential mechanisms remain elusive. In addition, it has been recently reported that the gut microbiota can be perturbed by microplastic exposure. For instance, microplastics, including polystyrene and polyethylene particles, could affect gut microbial composition and induce adverse effects such as inflammation [25,26]. Likewise, the profile of metabolites such as bile acids in the gut microbiota could also be impacted upon exposure to microplastics [27,28]. The gut microbiota not only directly impacts intestinal homeostasis locally through microbial metabolic products, but it also triggers systemic effects on remote tissues/organs such as liver, adipose, or brain by producing metabolites that can act as signaling molecules [29,30]. Moreover, the role of the gut microbiota in chemical toxicity has been well recognized [18]. Given the profound role of the gut microbiome in human health coupled with extensive and constant exposure to microplastics, it is of necessity to elucidate effects of microplastic exposure on the gut microbiome.

The main goal of the present study is to elucidate the effect of MP exposure on the gut microbiota. Microplastics comprise a variety of particles, of which polystyrene is the most classic example of a microplastic that exists widely throughout the environment. Provided the ubiquitous existence of polystyrene coupled with the use of polystyrene as a representative microplastic in a number of studies [26,27,28,31,32], polystyrene (5 μm, 0.1 mg/day) microplastic particles were used in the present study as representative microplastics for exposure to the gut microbiota. Polystyrene is widely used as a raw material in a variety of consumer products, including food packaging, leading to its ubiquitous environmental occurrence [33]. Thus, microplastic polystyrene is often used in a number of studies to investigate microplastic toxicity [26,27,34]. MP at 5 μm is the smallest diameter of plastic debris found in marine habitats [35], and it is within the smaller size range of particles that have been known to be ingested by aquatic organisms [36]. The dose is chosen based on environmentally relevant concentrations of MP in the environment [37]. Two complementary omic approaches have been employed to achieve a comprehensive understanding of how the gut microbiome is impacted by MP exposure at an environmentally relevant level. The 16S rRNA gene sequencing technique has been used to identify bacteria at the species level perturbed by MP exposure, which has been used as a mainstay of sequence-based bacterial analysis for decades. The other technique, activity-based metabolomics, including both untargeted and targeted approaches, allows the identification of metabolites with significant change upon exposure. An untargeted approach enables the comprehensive comparison of metabolomes under different conditions, while a targeted approach can enable the highly sensitive analysis of specific gut-microbiome-derived metabolites, which is critical in understanding drivers of physiological activities related to the gut microbiome.

## 2. Materials and Methods

### 2.1. Chemicals

Five-micrometer green, fluorescent polystyrene microplastic particles were purchased from Tianjin BaseLine ChromTech Research Center (Tianjin, China), with an excitation wavelength of 488 nm and an emission wavelength of 520 nm.

### 2.2. Animals and Experimental Design

Twenty male specific-pathogen-free (SPF) C57BL/6 mice (~4 weeks old) were purchased from SLAC Laboratory Animal Co., Ltd (Shanghai, China). After 1 week of acclimation, mice were randomly assigned into 6 cages with 3 or 4 mice per cage. A total of 10 mice were marked as controls (Control group, *n* = 10), and the rest of them were marked as MP-treated mice (treatment group, *n* = 10). The mice were housed under a temperature of 22 °C and 40–70% humidity with a 12/12 h light/dark cycle. Mice from the treatment group were given MP particles (0.1 mg/day [27,37]) by oral gavage; the control mice were given equivalent volume of water also by oral gavage. After 6-week MP treatment, fecal samples were collected individually, and these were put into liquid nitrogen immediately and stored at −80 °C for further experiments. Before sacrifice, mice were euthanized in a carbon dioxide chamber after 12 h of fasting. Mouse tissues were collected and quickly frozen in liquid nitrogen and stored at −80 °C. The animal experiment was conducted at Laboratory Animal Research Center of Zhejiang Chinese Medical University with approval by the Animal Care and Use Committee of Zhejiang Chinese Medical University (No. 20201103-08). All experiments were in accordance with relevant guidelines. All mice were treated humanely with regard for alleviation of suffering.

### 2.3. 16S rRNA Gene Sequencing

Microbial DNA was extracted from mouse fecal samples (*n* = 8 per group, ~50 mg per mouse) using QIAamp DNA Stool Mini Kit (Qiagen, Germany), as per the manufacturer’s instructions. Purified amplicons were pooled in equimolar and then paired-end sequenced using an Illumina MiSeq platform (Illumina, San Diego, CA, USA), as per the standard protocols by Majorbio Bio-Pharm Technology Co. Ltd. (Shanghai, China). The V4 region of the 16S rRNA gene was amplified by PCR with primers of 515 (5′-GTGCCAGCMGCCGCGGTAA) and 806 (5′- GGACTACHVGGGTWTCTAAT). PCR reaction conditions were 3 min at 95 °C, followed by 30 cycles of 45 s at 95 °C, 60 s at 50 °C and 90 s at 72 °C. Raw fastq files were quality-filtered by Trimmomatic and merged by FLASH. The taxonomy of each 16S rRNA gene sequence was analyzed by the RDP Classifier algorithm. OTUs were clustered using a 97% similarity cutoff with UPARSE (version 7.1). Tax4Fun, an open-source R package for pathway prediction, was used to profile functional genes of the gut microbiome, based on marker genes from the 16S rRNA gene sequencing data combined with the KEGG database of reference genomes [38].

### 2.4. Quantification of Fecal Short-Chain Fatty Acids (SCFAs)

Briefly, 50 mg of fecal samples (*n* = 6 per group) were mixed with an NaOH solution containing internal standards of SCFAs (acetate, butyrate, and propionate), and then homogenized for 2 min. The homogenates were centrifuged, and the supernatants were mixed with 1-propanol/pyridine (3:2, *v*/*v*) and propyl chloroformate for derivatization. Derivatives were extracted using hexane and transferred to an autosampler vial for injection. An Agilent 7820A GC-5977B MS system (Agilent Technologies, Santa Clara, CA, USA) using an HP-5 ms capillary column was applied for GC/MS analysis. The initial oven temperature was held at 50 °C for 2 min, then ramped to 70 °C at a rate of 10 °C min^−^^1^, to 85 °C at a rate of 3 °C min^−^^1^, to 110 °C at a rate of 5 °C min^−^^1^, to 290 °C at a rate of 30 °C min^−^^1^, and finally held at 290 °C for 8 min. In addition, the temperatures of the front inlet, transfer line, and electron impact (EI) ion source were set to 260, 290, and 230 °C, respectively [39].

### 2.5. Untargeted Metabolomic Analysis

To extract fecal metabolites, 50 mg of mouse fecal samples (*n* = 6 per group) were mixed with pre-cooled solution (methanol:acetonitrile:water = 2:2:1, *v*/*v*), which was added to 20 μL of L-2-chlorophenylalanine as the internal standard. Samples were extracted ultrasonically for 30 min at 5 °C. After 10 min of incubation at −20 °C, the sample was then centrifuged at 14,000× *g*/4 °C for 20 min. The supernatant was dried up using nitrogen. Before injection, the samples were dissolved with 100 μL of acetonitrile solution (acetonitrile:water = 1:1, *v*/*v*), followed by vortexing for 30 s and ultrasonic extraction for 5 min at 4 °C. The supernatant was used for injection after centrifuging at 14,000× *g*/4 °C for 15 min. Fecal metabolite profiles were analyzed using an Agilent 1290 Infinity LC Ultra High-Performance Liquid Chromatography System (UPLC) with a Hydrophilic Interaction Liquid Chromatography (HILIC) column in ShangHai Applied Protein Technology Co. Ltd. (Shanghai, China). A total of 20 µL of supernatant from each sample was mixed together as the quality-control sample. The mobile phase composition of A was as follows: water + 25 mM ammonium acetate + 25 mM ammonia. For B, it was as follows: Acetonitrile. The solvent gradients were as follows: at 0–0.5 min, B was 95%; at 0.5–7 min, B changed linearly from 95% to 65%; at 7–8 min, B changed linearly from 65% to 40%; at 8–9 min, B was maintained at 40%; at 9–9.1 min, B changed linearly from 40% to 95%; at 9.1–12 min, B was maintained at 95%. Mass spectrometry settings were as follows: ion source gas1: 60; ion source gas2: 60; curtain gas: 30; source temperature: 600 °C; ion sapary voltage floating: ±5500 V; TOF MS scan m/z range: 60–1000 Da; product ion scan m/z range: 25–1000 Da; TOF MS scan accumulation time: 0.20 s/spectra; product ion scan accumulation time: 0.05 s/spectra; and information-dependent acquisition (IDA) was applied for MS/MS with a collision energy of 35 ± 15 eV. Raw data, converted from Wiff format into mzXML format by ProteoWizard, were imported into XCMS software for baseline filtering, peak identification, retention time correction, and peak alignment. Identification of metabolites was performed based on the accuracy m/z value (<25 ppm) and the comparison of MS/MS spectra with an in-house database established by ShangHai Applied Protein Technology Co. Ltd. (Shanghai, China), which was built using available authentic standards.

### 2.6. Statistical Analysis

Differences in gut bacterial abundances were assessed by a nonparametric test via Metastats. Two-tailed Welch’s *t*-test was used to analyze metabolites that differed in abundance between groups corrected for the FDR. In addition, alpha rarefaction and principal co-ordinates analysis (PCoA) were used to assess diversities in the gut microbial communities. Principal components analysis (PCA) and a hierarchical clustering algorithm were used to visualize the comparison of metabolite profiles and pathways. The correlation matrix between gut bacterial species and metabolites was generated using Spearman’s correlation coefficient. Unless otherwise indicated, all results are expressed as mean values with standard deviation (** *p* < 0.01; * *p* < 0.05). The symbol * represents statistically significant difference.

## 3. Results

### 3.1. MP-Induced Alterations in Diversity and Composition of the Gut Microbial Community

After 6-week exposure of MP, there was no significant change in final body weight, weight gain, or food intake observed between the control and the MP-treated groups. Details regarding body weight, food intake, and organ indices are provided in the supplementary information (Appendix A). To investigate the impact of MP on the gut microbiome, fecal samples of mice were collected for taxonomic characterization and metabolite profiling after 6-week exposure to MP (Figure 1A). The overall composition of the gut microbial community was assessed by examining taxonomic similarity between the sequencing samples in the control and the MP-treated groups. As shown in Figure 1B, the sequences assigned to Firmicutes were more enriched in fecal samples from the MP-treated mice, whereas reads assigned to Bacteroidetes were slightly lower in these samples. Overall, the MP-treated group and the control group shared 170 bacterial species (Figure 1C). Interestingly, there were 12 and 19 unique genera in the gut microbiota of the control and MP-treated mice, respectively. Bacterial communities were clustered using Principal Coordinate Analysis (PCoA), which revealed that the gut microbiota of the MP-treated mice was distinctly clustered compared to that of the controls (Figure 1D).

Ace and Chao1 are indices estimating the number of OTUs in the community. A larger value of Ace or Chao1 indicates more OTUs of the microbiota community. Significantly increased values of Ace and Chao1 were observed in the MP-treated group compared to the controls (Figure 1E,F). Shannon and Simpson are indicators reflecting the diversity of the microbial community. Increased diversity comes with a larger Shannon value or a smaller Simpson value. MP exposure significantly increased the diversity of the gut microbiota community (Figure 1G). Taken together, we found that MP treatment increased the number of OTUs and diversity in the gut microbiota of mice. In addition, the phylogenetic tree shown in Figure 2A indicated that the gut microbiota were significantly modified by MP treatment. The red color represents bacteria that are abundant in the control mice, and the blue color represents abundant bacteria in the MP-treated group. As shown, phyla of Firmicutes and Campilobacterota were enriched in the mice upon microplastic exposure. LEfSe analysis disclosed that MP treatment is associated with an expansion of Clostridia and Lachnospiraceae (o_Lachnospirales) (Figure 2B). Particularly, at a species level, MP treatment is associated with an increase in a species from f_Muribaculaceae and a species from g_Anaerotruncus but a decrease in a species from g_Dubosiella (Figure 2C).

### 3.2. MP-Induced Changes in Functional Pathways of the Gut Microbiome

We next investigated changes in functional pathways of the gut microbiome in response to MP exposure. As shown in Figure 3A, the heatmap representing distribution of gene abundances shows strong comparison of functional pathways between the control and MP-treated groups. In particular, abundances of bacterial genes that were involved in pathways including xenobiotic biodegradation and metabolism, bacterial infectious disease, and drug resistance were significantly enriched in the MP-treated group compared to the controls (Figure 3B,D). In addition, abundances of bacterial genes encoding xenobiotic-related genes including ABC transporters, the bacterial secretion system, lysosomes, flagellar assembly, and quorum sensing were significantly perturbed with the administration of MP (Figure 3E–I).

### 3.3. MP Alters Metabolite Profiles of the Gut Microbiome

To further assess the impact of MP on metabolic products of the gut bacteria, an untargeted metabolomics experiment was performed on mouse fecal samples. Table 1 lists annotated dysregulated features with fold changes of ≥1.5 or ≤0.67 (*p* < 0.05) for microplastic challenges, including numerous metabolites that were related to gut microbial activities. PCA analysis clearly separated the two cultivars under both positive and negative ion modes (Figure 4A,C). The distribution of metabolic profiles of the MP-treated group was well-separated from the control group. Differential metabolites were screened with criteria of fold change >1.5 or <0.67 and *p*-Value < 0.05, which were visually represented with volcano plots (Figure 4B,D). In addition, enrichment analysis indicated that the pathways of cholesterol metabolism, primary and secondary bile acid biosynthesis, and taurine and hypotaurine metabolism, which play a key role in bile acid metabolism, were significantly enriched in the MP-treated mice (Appendix A). Fold change and classification of identified metabolites under positive and negative ion modes were visually presented in Figure 4E,F, respectively.

### 3.4. Key Metabolites That Are Associated with MP Exposure

The structures of altered metabolites are diverse, with a number of the metabolites being either directly generated or modified by the gut microbes. As shown in Figure 5, several typical gut-microbiota-related metabolites were differentiated upon MP exposure. Specifically, significantly increased levels of bile acids were observed in the gut microbiota (Figure 5A). Accordingly, the abundance of the bile acid metabolism pathway also showed significant increase upon MP treatment (Figure 5B). Moreover, modulation of purine and pyrimidine nucleoside abundances, in addition to pathways of nucleotide metabolism, was found in the gut microbiome of mice (Figure 5C,D). Given the essential role of SCFAs in metabolic activities of the gut microbiota, we additionally measured fecal levels of SCFAs. Fecal levels of SCFAs, including acetic acid, propionic acid, butyric acid, and isobutyric acid, were significantly decreased upon MP exposure (Figure 5F). Taken together, significant changes of key metabolites derived from the gut microbiota upon MP exposure were observed, which could contribute to the toxicity of MP.

### 3.5. Correlation between the Gut Microbiome and Metabolites

To explore the functional correlation between the gut microbiome changes and metabolite perturbations induced by MP treatment, we performed functional correlation analysis between the gut microbial species and metabolites with significant changes. Strong correlations were identified between the relative abundances of gut bacterial species and altered metabolite profiles (rho > 0.7 or <−0.7; *p* < 0.05). Figure 6 lists several typical gut-microbiota-related metabolites that are highly correlated with specific gut bacteria. For example, taurocholate, a key bile acid, negatively correlates with B1 (s__uncultured_bacterium_g__Dubosiella) but positively correlates with B2 (s__uncultured_bacterium_g__Anaerotruncus).

## 4. Discussion

We used high-throughput 16S rRNA gene sequencing and metabolomics profiling to investigate the impact of MP exposure on the gut microbiota. The results clearly showed that MP exposure induced a significant alteration in the gut microbial composition of mice. In addition, perturbations in gut bacterial composition were associated with changes in a variety of gut-microbiota-related metabolic products, suggesting that MP exposure not only perturbs the gut microbiota at the abundance level but also essentially changes the metabolite profile. Specifically, MP exposure significantly perturbed aspects of the gut microbiota, including its composition, diversity, and functional pathways that are involved in xenobiotic metabolism. A distinct metabolite profile was observed, which probably resulted from changes in gut bacterial composition and metabolic pathways induced by MP exposure. A number of key metabolites, including bile acids, purine and pyrimidine nucleosides, lipids, and SCFAs that are associated with MP exposure, may contribute to the toxic effects of MP.

The underlying mechanisms by which MP may exert toxic effects are still elusive. Mounting evidence suggests that metabolic changes associated with gut microbiome perturbations are key risk factors for the onset and development of various adverse outcomes [40]. The gut microbiome could not only directly impact intestinal homeostasis locally through bacterial products, but it could also trigger systemic effects on remote tissues including liver, adipose, or brain by producing metabolites that serve as signaling molecules [29,30]. Therefore, metabolic changes, especially perturbations in microbiota-generated metabolites, play an essential role in the disease development.

The gut microIiome is critical to the energy metabolism of the host. Dysbiosis of the gut microbiome may be associated with obesity and diabetes [41,42]. In the present study, we observed an increase in the proportion of Firmicutes and a slight decrease in the proportion of Bacteroidetes in the MP-treated mice compared to the controls, which is a typical characteristic of obesity-driven dysbiosis and is consistent with previous reports [6]. Similar changes regarding Firmicutes and Bacteroidetes were also reported in a previous study [27]. However, this result is in contradiction with another previous study, which reported a decrease in the proportion of Firmicutes and a slight increase in the proportion of Bacteroidetes in the MP-treated mice compared to the controls [28]. In addition, we observed an increase in microbial diversity in the MP-treated mice compared to the controls. Our follow-up results shown in Figure 3 and Appendix A indicate that MP may lead to the appearance of potentially pathogenic bacteria, providing a possible explanation for the increase of microbial diversity. Moreover, abundances of bacterial genes that are involved in pathways including bacterial infectious disease and drug resistance were significantly enriched in the MP-treated group compared to the controls (Figure 3C,D), further supporting the appearance of potentially pathogenic bacteria in the gut of the MP-treated mice. It is worth noting that the microbes in MP dilutes may also impact the composition of the gut microbiota of the MP-treated mice, and verification of sterility is able to exclude the possible effects of this factor.

The gut microbiome has profound roles in xenobiotic metabolism and evolved in the toxicity of environmental agents. Our results show enrichment of pathways involved in xenobiotic biodegradation and metabolism (Figure 3B). Particularly, abundances of bacterial genes encoding xenobiotic-related genes including ABC transporters, the bacterial secretion system, and lysosomes were significantly altered in the gut microbiome of the MP-treated mice. It is reported that ABC transporters are capable of exporting microplastic particles [43]. Consistently, we also observed alterations in metabolites that are related to ABC transporters (Appendix A). Therefore, MP treatment would induce functional changes in xenobiotic-metabolism-related pathways of the gut microbiome, providing additional evidence regarding the interactions between the gut microbiome and xenobiotic metabolism and toxicity.

Recent evidence suggested that perturbations of the gut microbiome and its functions may be a potential mechanism underlying the toxic effects of environmental agents [18]. The gut bacteria could directly communicate with the host through the production of a variety of endogenous metabolites. Bile acids are cholesterol derivatives that are synthesized in the liver before undergoing extensive enterohepatic recycling as well as modification by gut bacteria. It is established that not only are bile acids involved in digestion and absorption, but also they act as signaling molecules affecting diverse pathways by activating a number of nuclear receptors [44]. Bile acids and intermediates, in addition to bile-acid-related pathways, were significantly perturbed in the MP-treated mice, indicating that MP exposure affects the homeostasis of bile acids. The underlying mechanisms remain elusive; however, MP-induced gut-microbiome perturbations may be involved. Previous reports demonstrated that the gut microbiota would affect primary and secondary bile acid profiles in the tissues of antibiotic-treated rats [45]. Moreover, it is reported that bile-acid signaling via the relevant receptors is associated with the regulation of the immune system and inflammatory response [46]. It is particularly interesting that taurocholate, a key bile acid molecule, negatively correlates with abundances of s__uncultured_bacterium_g__Dubosiella, whereas it positively correlates with abundances of s__uncultured_bacterium_g__Anaerotruncus. Therefore, alterations in the bile-acid profile in the MP-treated mice could be associated with species changes in the gut microbiota, suggesting the involvement of the bacterial metabolic activities of bile acids in chemical toxicity. MP-induced alterations in bile-acid-related metabolites were reported in a previous study; their results showed that exposure to 5 µm MP impacted bile acid metabolism of mice via measurement of the total bile acids in serum and the liver [27]. This is further confirmed in a more recent study in which plasma metabolites were profiled in mice upon MP exposure. Perturbations of bile acids and derivatives were also observed in mouse plasma such as taurochenodeoxycholate (TUDCA) [47], which increased with a fold change of two in fecal samples of the MP-treated mice compared to that of the controls, according to the results of the present study. It is worth noting that we cannot exclude an effect on host metabolism due to styrene monomers derived by MP degradation, which could also exert an impact on biliary and taurine metabolism. Likewise, an important class of gut-microbiota-derived metabolites, the functions of SCFAs have been extensively studied. For example, it is well documented that butyrate has anti-inflammatory effects, mainly via suppression of nuclear factor kappa β (NF-κB) activation in macrophages and histone deacetylation (HDAc) in acute myeloid leukemia [48,49]. It has recently been reported that propionate and butyrate have a potential role in regulatory T-cell production and function at the whole-animal level through inhibition of HDAc [50,51]. Thus, perturbations of metabolic profiles, especially bile acids and SCFAs in the gut microbiome, may be linked to MP-induced toxic effects. Taken together, these results support the hypothesis that perturbations in the gut microbial composition and key metabolites could be one of the underlying mechanisms of MP toxicity.

## 5. Conclusions

In summary, MP exposure perturbs the gut microbial composition and key metabolites in mice. This finding represents an important step toward understanding how MP exposure affects the gut microbiome and its functions. Future studies are warranted to address more intriguing issues. For instance, the dose-, time-, and particle-size-dependent effects of MP exposure on the gut microbiota need to be defined. Nevertheless, our results indicate that MP exposure not only changes the gut microbiota at the abundance level but also substantially alters the metabolic profiles with perturbations in key metabolites. Key metabolites and bacterial species were identified. Perturbations in the gut microbiota and these gut-microbiota-related metabolites by MP exposure support the hypothesis that environmental chemicals including MP could lead to toxic effects via perturbation of the gut microbiome and its metabolic profiles.

## Figures and Tables

**Figure 1 metabolites-13-00530-f001:**
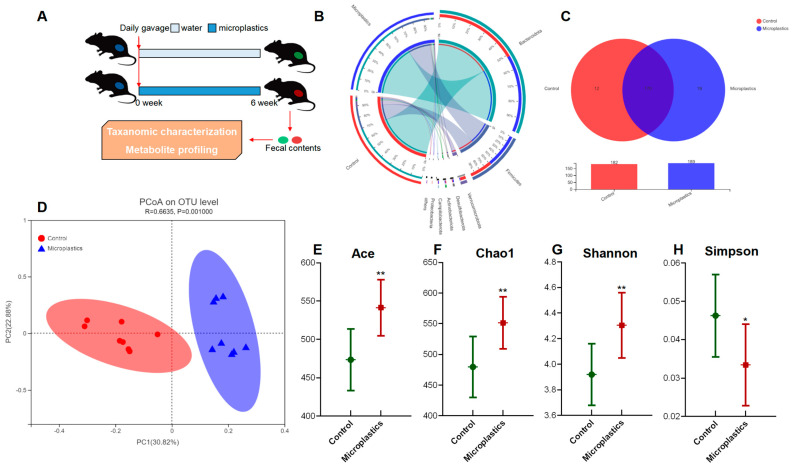
MP-induced alterations in the mouse gut microbial community. (**A**) Experimental design. (**B**) Circos diagram for component profiles of the gut microbial community at phylum level (*n* = 8). (**C**) Venn diagram for component profiles of the gut microbial community at species level (*n* = 8). (**D**) Component profiles analyzed by PcoA model (*n* = 8). (**E**–**H**) Alpha diversity indices of Ace, Chao1, Shannon, and Simpson (*n* = 8, * *p* < 0.05, ** *p* < 0.01).

**Figure 2 metabolites-13-00530-f002:**
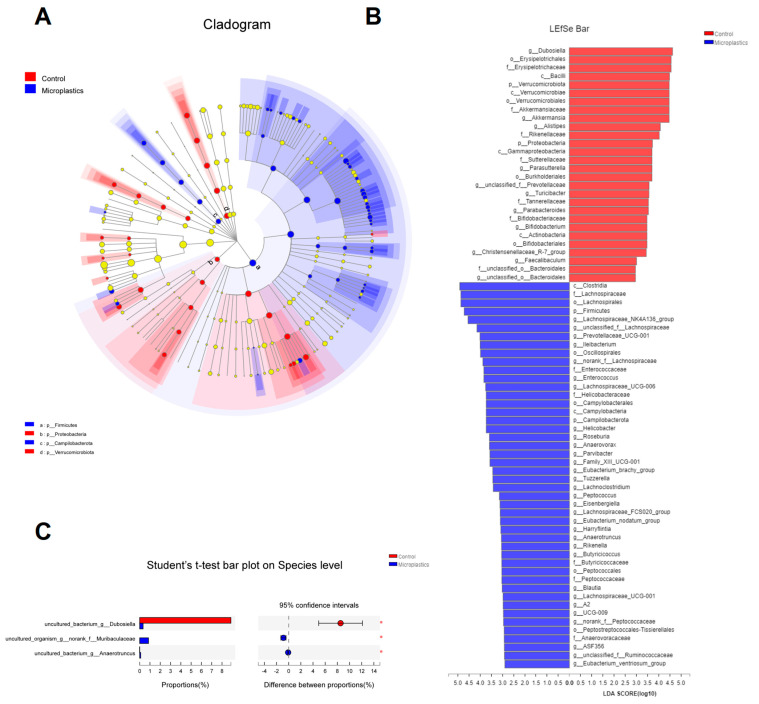
(**A**) Cladogram (phylum to genus) by discriminant analysis of Lefse with different colors representing control (Red) or microplastics (Blue) groups (*n* = 8). (**B**) Discriminant analysis of Lefse multi-level species difference (*n* = 8). (**C**) Community barplot analysis at species level (*n* = 8, * *p* < 0.05).

**Figure 3 metabolites-13-00530-f003:**
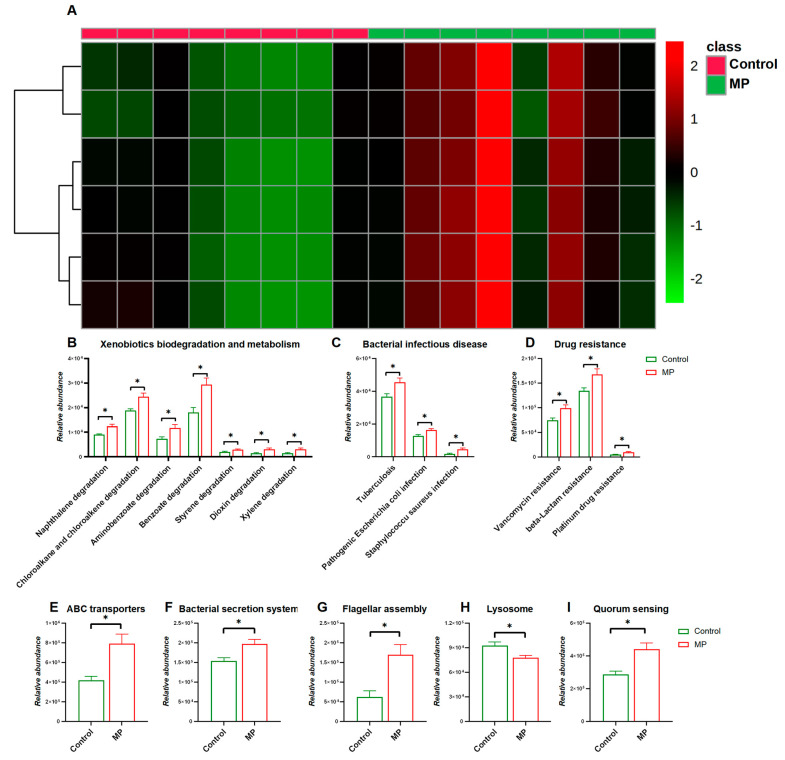
Comparisons of Functional pathways (*n* = 8). (**A**) Heatmap constructed by abundances of bacterial pathways at KEGG Level 1. The abundance of pathways related to xenobiotic biodegradation and metabolism (**B**), bacterial infectious disease (**C**), and drug resistance (**D**). The abundance of pathways related to ABC transporters I, bacterial secretion system (**F**), flagellar assembly (**G**), lysosome (**H**), and quorum sensing (**I**). *, *p* < 0.05.

**Figure 4 metabolites-13-00530-f004:**
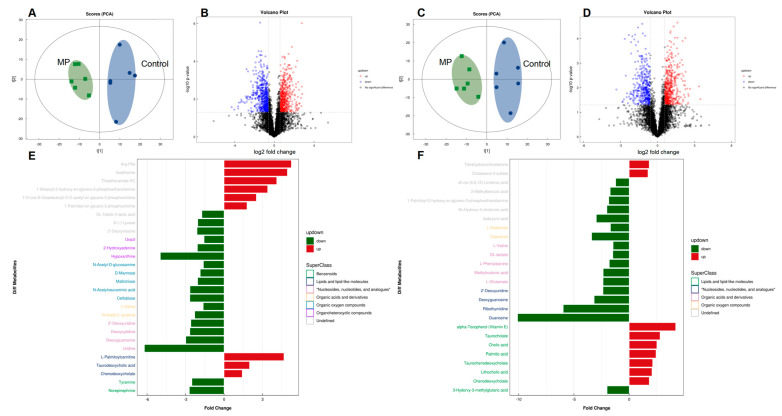
Comparisons of metabolite profiles (*n* = 6). Volcano plots represent distribution of metabolite fingerprints in positive (**A**) and negative (**C**) ion modes. PCA score charts of metabolite profiles in positive (**B**) and negative (**D**) ion modes. Fold change and classification of identified metabolites under positive (**E**) and negative (**F**) ion modes.

**Figure 5 metabolites-13-00530-f005:**
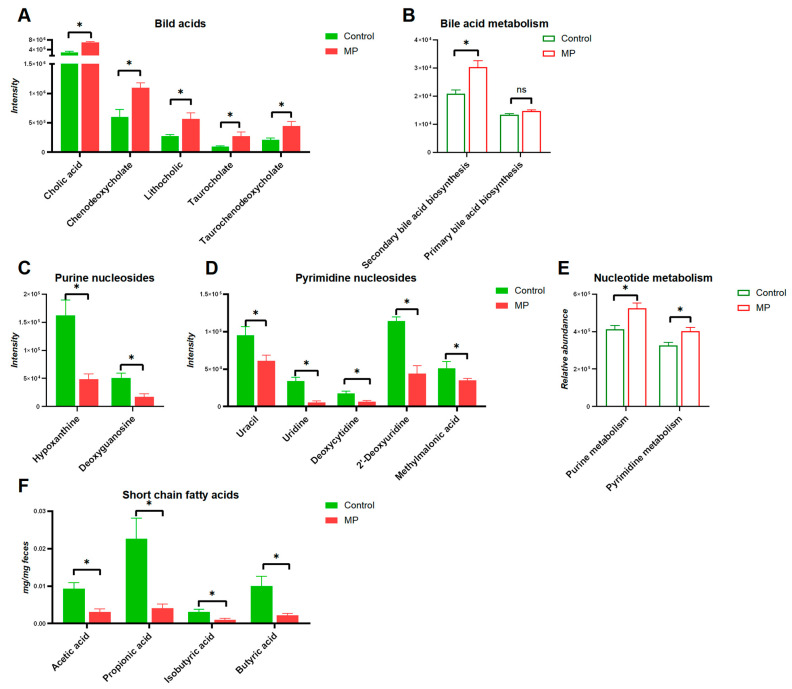
Key metabolites in the gut microbiota associated with MP exposure. (**A**) bile acids (*n* = 6). (**B**) Pathways involved in bile acid metabolism (*n* = 8). (**C**) Purine nucleosides (*n* = 6). (**D**) Pyrimidine nucleosides (*n* = 6). (**E**) Pathways involved in nucleotide metabolism (*n* = 8). (**F**) Short-chain fatty acids (*n* = 6). *, *p* < 0.05.

**Figure 6 metabolites-13-00530-f006:**
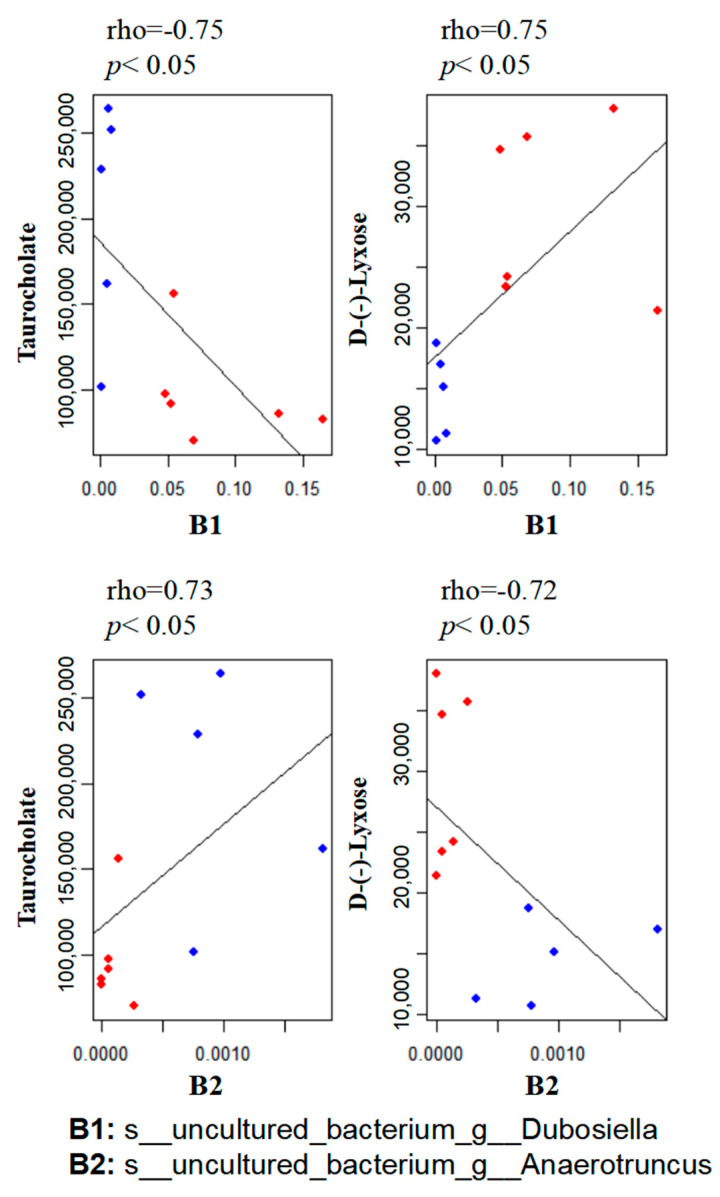
Scatter plots illustrating statistical associations between key metabolites and gut bacterial species (Blue dot: MP; Red dot: Control; rho > 0.7 or <−0.7; *p* < 0.05).

**Table 1 metabolites-13-00530-t001:** Significantly dysregulated (*p* < 0.05) metabolomic classes and the corresponding metabolites in mice exposed to microplastic.

Annotated Feature	Adduct	m/z	rt(s) ^a^	VIP ^b^	Fold Change ^c^	*p*-Value	HMDB ID
Nucleosides, nucleotides, and analogues							
2’-Deoxyuridine	(M − H)−	227.0671	112.8	10.7	0.4	6.80 × 10^−4^	12
Deoxycytidine	(M + H)+	228.0964	206.6	1	0.4	7.24 × 10^−3^	14
Deoxyguanosine	(M − H)−	266.0890	230.3	2.1	0.3	6.30 × 10^−3^	85
Guanosine	(M − H)−	282.0840	262.8	1.3	0.1	1.20 × 10^−2^	133
Uridine	(M + H)+	245.0757	159.2	1.7	0.2	3.44 × 10^−4^	296
Ribothymidine	(M − H)−	257.0780	142	4	0.2	3.20 × 10^−3^	884
Lipids and lipid-like molecules							
3-Hydorxy−3-methylglutaric acid	(M − H)−	161.0452	373.8	1.1	0.5	9.00 × 10^−4^	355
alpha-Tocopherol (Vitamin E)	(M − H)−	429.3724	31.5	1.9	4.2	3.10 × 10^−2^	1893
Cholic acid	(M − H)−	407.2802	227.2	14	2.5	1.70 × 10^−4^	619
Palmitic acid	(M − H)−	255.2327	46.7	9.4	2.4	4.60 × 10^−2^	220
Taurochenodeoxycholate	(M − H)−	498.2886	140.7	3	2.1	1.60 × 10^−2^	951
Taurocholate	(M − H)−	514.2840	200.3	2.4	2.8	4.40 × 10^−2^	36
Chenodeoxycholate	(M + CH_3_COO)−	451.3053	160.3	4.4	1.8	8.00 × 10^−3^	518
Lithocholic acid	(M + CH_3_COO)−	435.3105	82.6	3.3	2.1	2.10 × 10^−2^	761
L-Palmitoylcarnitine	(M + H)+	400.3401	172.9	2.4	4.7	1.41 × 10^−2^	222
Taurodeoxycholic acid	(M + NH4)+	517.3270	140.8	1.7	2	1.01 × 10^−2^	896
Organic acids and derivatives							
DL-lactate	(M − H)−	89.0243	304.1	1.8	0.7	4.00 × 10^−2^	1311
L-Glutamate	(M − H)−	146.0458	398.3	2	0.4	4.70 × 10^−2^	148
L-Phenylalanine	(M − H)−	164.0719	261.5	3.1	0.6	5.00 × 10^−3^	159
L-Valine	(M − H)−	116.0714	304.9	2.3	0.7	2.50 × 10^−2^	883
Methylmalonic acid	(M − H)−	117.0188	104.9	1.2	0.4	1.10 × 10^−2^	202
L-Arabinose	(M − H)−	149.0449	133	2	0.6	1.50 × 10^−2^	646
Galactinol	(M + CH_3_COO)−	401.1292	391.2	1.1	0.3	1.20 × 10^−2^	5826
N-Acetyl-L-tyrosine	(M − H + 2Na)+	268.0606	242.4	1.3	0.4	3.45 × 10^−2^	866
N-Acetyl-D-glucosamine	(M + H)+	222.0966	256.4	3	0.6	2.76 × 10^−2^	215
N-Acetylneuraminic acid	(M + H)+	310.1121	373.6	2.2	0.4	4.82 × 10^−5^	230
Cellobiose	(M + NH_4_)+	360.1487	389.6	4.4	0.4	3.05 × 10^−2^	55
D-Mannose	(M + NH_4_)+	198.0958	302.8	2	0.5	4.46 × 10^−3^	169
Maltotriose	(M + NH_4_)+	522.2001	449.6	1.6	0.5	2.10 × 10^−2^	1262
Benzenoids							
Tyramine	(M + H)+	138.0900	218.2	1.2	0.4	2.46 × 10^−3^	306
Norepinephrine	(M + H − H_2_O)+	152.0691	105.2	2.7	0.4	3.43 × 10^−4^	216
Organoheterocyclic compounds							
2-Hydroxyadenine	(M + H)+	152.0559	262.4	1.5	0.5	3.03 × 10^−2^	403
Hypoxanthine	(M + H)+	137.0448	217.3	4.2	0.2	4.17 × 10^−2^	157
Uracil	(M + H)+	113.0334	84.7	1.9	0.6	3.31 × 10^−2^	300
Others							
1-Palmitoyl−2-hydroxy-sn-glycero−3-phosphoethanolamine	(M − H)−	452.2773	200.6	4.6	0.5	2.00 × 10^−2^	n.a.^d^
2-Methylbenzoic acid	(M − H)−	135.0445	133.3	1.2	0.6	1.20 × 10^−2^	2340
3b-Hydroxy−5-cholenoic acid	(M − H)−	373.2733	61.2	2.5	0.5	3.00 × 10^−2^	308
gamma-Linolenic acid	(M − H)−	277.2174	46.1	6.4	0.8	4.20 × 10^−2^	3073
Cholesterol 3-sulfate	(M − H)−	465.3042	26.3	15.8	1.7	1.40 × 10^−2^	653
Isobutyric acid	(M − H)−	87.0452	132.6	6	0.3	1.30 × 10^−4^	1873
Tetrahydrocorticosterone	(M − H)−	349.2373	67.6	1.4	1.8	7.50 × 10^−3^	268
Isoetharine	(M + CH_3_CN + Na)+	303.1689	197.6	1.6	4.9	2.00 ×10^−2^	14366
1-Palmitoyl-sn-glycero−3-phosphocholine	(M + H)+	496.3396	194.5	10.3	1.8	3.93 ×10^−3^	n.a.
1-Stearoyl−2-hydroxy-sn-glycero−3-phosphoethanolamine	(M + H)+	482.3230	195.7	3.1	3.4	1.73 ×10^−5^	n.a.
2’-Deoxyinosine	(M + H)+	253.0924	179.6	3.5	0.5	3.21 ×10^−2^	71
Arg-Phe	(M + H)+	322.1851	340.8	1.2	5.3	1.28 ×10^−3^	n.a.
DL-Indole−3-lactic acid	(M + H − H_2_O)+	188.0693	259.1	1.3	0.6	3.25 ×10^−2^	671
Thioetheramide-PC	(M + Na)+	758.5646	129.4	2.1	4.1	5.77 ×10^−3^	n.a.
D-(-)-Lyxose	(M + NH_4_)+	168.0853	152.6	1.3	0.5	9.26 ×10^−4^	n.a.
1-O-(cis−9-Octadecenyl)−2-O-acetyl-sn-glycero−3-phosphocholine	(M + H)+	550.3832	188.7	1.7	2.5	4.49 ×10^−4^	n.a.

^a^ rt—retention time; ^b^ VIP—variable importance in projection; ^c^ the value of fold change higher than or equal to 1.5 indicates that the corresponding metabolite was upregulated, and the value of fold change lower than or equal to 0.7 indicates that the corresponding metabolite was downregulated; ^d^ n.a., the HMDB ID is not available for the corresponding metabolite.

## Data Availability

Not applicable.

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
