# Peer review of "Deciphering Gut Microbiome Responses upon Microplastic Exposure via Integrating Metagenomics and Activity-Based Metabolomics"

_metabolites, 2023, doi:10.3390/metabo13040530_

Round 1

Reviewer 1 Report

The main goal of the present study was to elucidate the effect of MP exposure on the gut microbiota. Two complementary omic approaches were employed to achieve a comprehensive understanding of how gut microbiome was impacted by MP exposure at environmentally relevant level. 16S rRNA gene sequencing technique was used to identify bacteria at the species level perturbed by MP exposure, while untargeted (GC/MS of short fatty acids) and targeted metabolomics approaches, were used for the identification of metabolites with significant change upon exposure.

The manuscript is well written and clear, however it needs to be improved in the description of methods and results. The discussion and conclusions must be better supported by the obtained results.

Major concerns:

1)      The authors should shortly describe the experimental conditions of GC/MS measurement or cite a previous paper using the same procedure.

2)      to move the p values from line 151 to line 157. To detail what they represent

3)      The number of mice were 10, but in figure 1D only eight subjects for group are present in the score plot. Why?

4)      The format pdf of the manuscript does not allow to read the text in the figures. The legends of all the figures should be more exhaustive and should better describe the single figures of the panels.

5)      Why in figure 4A and 4C the number of mice is six for group? What is the cause of this reduction of the number from 10 to 6 for each group? The number of the subjects is reduced from an analysis to another. The authors must explain.

6)      Table1: what does n.a. mean? Many metabolites indicated with n.a. on the contrary have a HMDB ID. The table 1 needs a better description in the legend.

7)      Figure 6: The linear correlations displayed in figure 6 B1, B2, B3 and B5 may be influenced by a subject that show high levels of those metabolites. Is it the same subject in all four correlation plots? The authors should remove that subject from analysis and repeat the test. What is the mean of this correlation test? These bacterial species are not varied in the MP group.

8)      The authors write in discussion section: “Of particular interest, taurocholate, a key bile acid molecule, positively correlates with 509 three MP-associated bacterial species. Therefore, alterations in bile acid profile in MP-treated mice could be derived from the gut microbiota, further supporting the involvement of bacterial metabolic activities of bile acids in chemical toxicity”. This sentence is not supported by results. The bacterial species that correlate with the biliary metabolites are not those varying with MP. treatment.

9)      The authors must better discuss the effects of MP on biliary and taurine metabolism, because the variations could not depend only on microbiota changes. The authors cannot exclude an effect on host metabolism due to styrene monomers derived by MP degradation.

10)  The authors have not related the changes observed by 16SrRNA gene with those obtained by targeted and untargeted metabolomics.

Reviewer 2 Report

This manuscript aimed to interpret the response of microplastic polystyrene exposure to intestinal microbiome through C57BL/6 mouse model combined with 16S rRNA high-throughput sequencing and metabonomics analysis.With the current explosion in plastic production leading to environmental pollution and microplastic contamination throughout the food chain, it is important for human health to study the effects of microplastic exposure on the human gut microbiome. Although the authors showed that microplastic polystyrene exposure significantly changed the gut microbiome, the specific changes were not analyzed, and the significance of the study was not highlighted.

Some comments are described below:

1. Line 36 The paragraph format is obviously incorrect.

2. The introduction can add these contents: an introduction to the effects of microbial plastic exposure on intestinal microbes; The specific reasons for using polystyrene are expounded; The mechanism of intestinal microbial metabolism.

3. The format of the references should be consistent with the format required by the journal.

4. Line 98 There is a space between the number and the temperature unit, but there is no space between the number and the temperature unit on Line 125. Please keep the format the same.

5. Some English abbreviations in the text can be explained or given full names, such as HILIC, PCA, etc.

6. The explanation of Part 3.1 of the Results is not specific. It can specifically describe which microflora changes after the addition of microbial plastics.

7. The explanation in part 3.2 of the Results is not specific. The cause and consequences of the significant changes in ABC transporters and bacterial secretion system, etc., are not explained.

8. Materials and methods of part 2.3 16S rRNA gene sequencing can specifically describe PCR primers and reaction conditions.

9. The scanning electron microscope results of the intestinal tract of mice before and after polystyrene exposure can be added to better understand the changes of intestinal microorganisms.

10. Line 224 “Fig 3B-D”, and Line 300 “Fig. 4A&C”, Write in a consistent format.

Reviewer 3 Report

Review Deciphering gut microbiome responses upon microplastic exposure via integrating metagenomics and activity-based metabolomics

The authors present work on the effects of microplastics on fecal microbiome and metabolome in mice. The manuscript would benefit by correction by a native English speaker. In addition, the manuscript suffers from unreadable figures (too small, too small fonts), missing details, and not up-to-date referencing. In detail:

How were the microplastic beads sterilized before gavage?

How much fecal pellet material was used for 16S microbiome sequencing?

Did the authors use 50 micrograms of fecal pellet or 50 mg of fecal pellet?
What amount was used for the untargeted metabolomics using LCMS?

Typically, water is avoided when doing GCMS, so did the authors inject directly from aqueous samples or is something missing from the protocol?

For GCMS, column and temperature profiles need to be included.

For LCMS, column and solvent gradients need to be included, as well as the settings on the mass spec.

Figure 1, panels B and C are not readable (too small fonts)

Figure 4, all panels are unreadable due to small font size.

References stop at 2020. Since then, lots of newer papers have been published on microplastics and the gut microbiome. Authors need to do a thorough review of recent literature and include papers like

Jiang, P., Yuan, G.H., Jiang, B.R., Zhang, J.Y., Wang, Y.Q., Lv, H.J., Zhang, Z., Wu, J.L., Wu, Q. and Li, L., 2021. Effects of microplastics (MPs) and tributyltin (TBT) alone and in combination on bile acids and gut microbiota crosstalk in mice. Ecotoxicology and environmental safety, 220, p.112345.

Round 2

Reviewer 1 Report

The manuscript has been improved.

Author Response

Review #1: The manuscript has been improved.

Response: We again thank the reviewer for your constructive comments that helped us refine and develop our manuscript. We truly appreciate your attentive review service.

Reviewer 2 Report

According to the last revision suggestion, the author has made a good revision. However, the following manuscripts revisions are needed:

1. Line 92 "was" should be "is".

2. Line95-103 tenses, take a closer look, some tenses are inconsistent.

3. Line 135 “90 s”, and Line 135 “30 seconds”,Be consistent in format.

4. Line 546 evolvesshould be evolved.

5. Line 397 “ (Fig. 5C&Fig. 5D)” ,and Line 543 “ (Fig. 3C-Fig. 3D)”,Be consistent in format.

Author Response

Reviewer #2: According to the last revision suggestion, the author has made a good revision. However, the following manuscripts revisions are needed:

Response: Many thanks to the reviewer for your attentive review. We have revised our manuscript again as per the reviewer’s kind suggestions. Detailed point-to-point responses are as below. Detailed revisions of the manuscript were marked in the file entitled manuscript with marked revisions. Line numbers indicated below refer to the manuscript file without marked changes.

  1. Line 92 "was" should be "is".

Response: Thanks, it is now fixed (Line 91).

  1. Line95-103 tenses, take a closer look, some tenses are inconsistent.

Response: Thanks a lot, we have corrected inconsistent tenses (Line 94-103).

  1. Line 135 “90 s”, and Line 135 “30 seconds”,Be consistent in format.

Response: Thanks, it is now fixed (highlighted, Line 161).

  1. Line 546 ”evolves“ should be “evolved”.

Response: Thanks, it is now fixed (highlighted, Line 478).

  1. Line 397 “ (Fig. 5C&Fig. 5D)” ,and Line 543 “ (Fig. 3C-Fig. 3D)”,Be consistent in format.

Response: Thanks a lot for pointing this out, we have revised it and it is now in a consistent format (highlighted, Line 472).

Reviewer 3 Report

Review 2

The authors address several concerns (materials/methods) and provide higher quality figures and images. However, some comments remain.

Distilled water is not bacteria-free. Although outside of the scope of this paper, it would be good to spread microplastics diluted in distilled water on nutrient agar plates to verify sterility.

The authors add references 25-30 in their introduction with recent work on microplastics, but they do not discuss these recent works in their discussion. Since in these and other papers mice are also exposed to microplastics and microbiomes (16S) and metabolomes are collected, the authors need to compare their data with others in their discussion to highlight similarities and differences. This needs to be addressed before publication.

Additional work that needs discussing:

Jing, J., Zhang, L., Han, L., Wang, J., Zhang, W., Liu, Z. and Gao, A., 2022. Polystyrene micro-/nanoplastics induced hematopoietic damages via the crosstalk of gut microbiota, metabolites, and cytokines. Environment International, 161, p.107131.

Author Response

Reviewer #3: The authors address several concerns (materials/methods) and provide higher quality figures and images. However, some comments remain.

Response: We are very grateful to the reviewer for your attentive review and constructive suggestions. We have carefully addressed your concerns and revised the manuscript according to your kind suggestions. Detailed point-to-point responses are as below. Detailed revisions of the manuscript were marked in the file entitled manuscript with marked revisions. Line numbers indicated below refer to the manuscript file without marked changes.

Distilled water is not bacteria-free. Although outside of the scope of this paper, it would be good to spread microplastics diluted in distilled water on nutrient agar plates to verify sterility.

Response: We appreciate the reviewer’s kind suggestion, and we agree with the reviewer that verification of sterility is very important if the experiment requires sterility. The present study does not have such requirement. Nevertheless, we added discussion of limitation regarding this point, and thanks again for the reviewer’s kind reminder (Line 474-476).

The authors add references 25-30 in their introduction with recent work on microplastics, but they do not discuss these recent works in their discussion. Since in these and other papers mice are also exposed to microplastics and microbiomes (16S) and metabolomes are collected, the authors need to compare their data with others in their discussion to highlight similarities and differences. This needs to be addressed before publication.

Response: Thanks a lot for pointing this out, we have added discussion to highlight similarities and differences between our results and others.

Line 459-466: similarities and differences regarding microbiomes (16S).

Line 510-517: similarities and differences regarding metabolomes.

Additional work that needs discussing:

Jing, J., Zhang, L., Han, L., Wang, J., Zhang, W., Liu, Z. and Gao, A., 2022. Polystyrene micro-/nanoplastics induced hematopoietic damages via the crosstalk of gut microbiota, metabolites, and cytokines. Environment International, 161, p.107131.

Response: Thanks a lot for pointing this out, we have added this work in our discussion (Line 515, Ref 47).